# Activity Videos Effect on Four-, Five- and Six-Year-Olds’ Physical Activity Level in Preschool

**DOI:** 10.3390/sports11030056

**Published:** 2023-02-28

**Authors:** Karin Kippe, Pål Lagestad

**Affiliations:** 1Department of Preschool Teacher Education, Queen Maud University College, Thrond Nergaards gt. 7, 7044 Trondheim, Norway; 2Department of Teacher Education, Nord University, Høyskolevegen 27, 7600 Levanger, Norway

**Keywords:** preschool, preschool children, physical activity videos, accelerometers

## Abstract

Physical activity provides positive health benefits for preschool children. The aim of this study is to examine the effect of physical activity videos on the physical activity levels of children aged 4, 5 and 6 in preschool time. Two preschools served as a control group, and four served as intervention groups. The study included 110 children aged 4–6 years, all wearing accelerometers in the preschool for two weeks. In the first week, both the control group and the intervention group carried out their ordinary activities. In the second week, the four preschools in the intervention group used the activity videos, while the control group continued their ordinary activities. The main finding is that the activity videos only increased the 4 year olds’ physical activity in MVPA (moderate to vigorous physical activity) from pre-test to post-test. Furthermore, the results show significantly increased CPM (counts per minute) in preschool among 4- and 6-year-old children in the interventions group from pre-test to post-test. However, the children in the control group did not have a significant change in their CPM or MVPA from pre-test to post-test. Our findings indicate that the use of activity videos may increase preschool children’s activity levels at preschool, but that the videos need to be developed differently depending on the age of the children.

## 1. Introduction

Research has documented that regular physical activity (PA) yields positive health gains for children. In a review study, Carson et al. [1] found that PA was connected to motor development, psychosocial health, bone and skeletal health, and cardiorespiratory health in children. Furthermore, PA reduces the risk of cardiovascular disease in children significantly [1,2,3]. From a national health perspective, it is therefore very important to survey the PA levels of the population. This forms the foundation on which to build strategies and action plans to stimulate increased PA in the population. The World Health Organization [4] recommends that children should have 60 min of moderate to vigorous physical activity (MVPA) daily. Dobell et al. [5] point to the fact that there is mixed evidence for children’s physical activity levels. Earlier research has shown, however, that not all children satisfy these health recommendations and that the PA level among some preschool children is too low [6,7,8,9,10].

Previous research highlights preschool as a strong determinant of children’s PA [11,12,13]. A study [7] demonstrated that physical activity level during preschool was the main contributor to preschool children’s physical activity level on weekdays and found a positive association between physical activity level at preschool and leisure time. The study indicated that preschools increase inequality according to physical activity level among preschool children, contributing to creating differences among low-active and high-active children. Preschools have an important role in promoting plans for PA that provide space and time for children to be physically active. According to Mitchell [14], school-based physical education is one option to effectively increase physical activity in children. In Norway, almost all children from 3 to 6 years old are in preschool most of their waking hours [15], and preschools have a major influence on children’s PA levels. Several studies show that children receive approximately 2/3 of their PA while they are in preschool [6,7,16]. Consequently, it is asserted that preschool can contribute to influencing children’s health, as well as to levelling social differences, which constitutes an important principle underlying public health work [17]. Furthermore, several studies show that lifestyle behaviour will follow the same trend from preschool age up to adulthood [18,19,20]. It is suggested that PA should be prompted within the first five years of life because children’s activity patterns are relatively more easily influenced and open to changes and adoptions during this developmental period [21,22].

Several researchers highlight the importance of preschool staff being involved and making efforts to promote children’s PA, and furthermore, that policy and practice in preschool greatly impact the total PA level of children [22,23,24]. According to Vanderloo et al. [25], a lack of stimulation or inactive role models will demotivate children’s participation in physically active play. A study by Karlsen and Lekhal [26] showed that the preschool staff participated in children’s play only 15.5 percent of the time the play took place. There was also little supportive action among the preschool staff, such as comments in the form of explanations, questions or conversations. There was also little help or concrete guidance from the preschool staff in the children’s play. In contrast, Karlsen and Lekhal [26] claim that the staff’s role is to provide opportunities that enrich and extend children’s own thinking. According to Hussain [27], the preschool staff play a central role in both limiting and enabling progress in physically active play, generating and maintaining the children’s interest in further exploration. This approach enables the staff to respond to the children’s interests and together create learning situations with children that are emerging and meaningful for all children. Furthermore, a study by Fossdal et al. [28] found a significant association between preschool employees’ average PA levels during preschool hours and children’s corresponding activity levels during preschool hours.

The study by Fossdal et al. [28] demonstrates the importance of active employees in preschool. Moreover, Fossdal et al. [28] found that children spent, on average, more minutes in MVPA during weekdays than on the weekends, and since most of the MVPA during weekdays was achieved in preschool, this finding indicates that preschool is an important arena for children’s daily physical activity. Several studies show that the facilitation and organization of PA at preschools increase children’s total daily PA level [29,30,31,32,33]. Pate et al. [31] found that organized PA, the facilitation of PA, and the integration of PA in school preparatory activities increased 4 year olds‘ PA levels in moderate to vigorous PA, but not total PA per day. This is in line with Annesi, Smith and Tennant [34], who found that organized activity increased 5 year olds‘ time in MVPA, while there were no differences in sedentary time compared to the control group. Organized activity is planned and structured activity where the employees facilitate the activity to include all children. It is relevant to point out the importance of preschool for children’s PA; for example, the study by Aguilar-Farias [35] shows that during the early stages of the COVID-19 pandemic, time spent in PA decreased, recreational screen time and sleep duration increased, and sleep quality declined among preschool children. Øvreås et al. [36] pointed out that the successful implementation of interventions with the aim of increasing children’s PA levels requires the inclusion of all employees. Therefore, the design of interventions in accordance with the preschool’s capacity and the employees’ competence can create lasting results.

Video physical activity involves PA games where children play remotely with another child over the internet [37]. The terms for physically active games include: “Active video game”, “Activity-promoting video game”, “exergame”, “Kinect”, “Playstation”, “Wii”, “Xbox”, and “Nintendo” [38]. Activity videos can counteract sedentary behaviour and increase participation in PA [39]. Furthermore, Chacón-Cuberos [40] point out that students have a positive attitude to physical activity using “exergames”. Peng et al. [41] highlight that playing active video games generally is equivalent to light-to-moderate PA among children. In addition, Ramirez-Granizo et al. [39] found that activity videos are not able to replace PA in MVPA. However, the study of Kooiman et al. [37] shows that exergaming among secondary students can affect the heart rate, corresponding to moderate intensity. Physical activity with an effect on the heart rate is important due to the fact that higher levels of physical activity and cardiorespiratory function are associated with better health and reduce the risk of cardiovascular disease in children significantly [1,2,3].

The previous discussion points to the importance of the preschools’ contribution to children’s PA level and the children’s opportunity to fulfill the health recommendations of 60 min of PA in MVPA daily. The aim of the study was to elucidate whether the use of activity videos showing children in PA had an effect on children’s PA in preschool time (MVPA and CPM) among 4-, 5- and 6 year old children compared to ordinary activities.

The study has the following research question:

To what extent does the use of PA videos influence 4-, 5- and 6 year olds’ PA levels (MVPA and CPM) in preschool compared to ordinary activity without the use of videos?

## 2. Materials and Methods

### 2.1. Subjects

Of 24 preschools in a municipality in the northern part of Troendelag, a county located in central Norway, six preschools were randomly selected to participate in the study, independently of the type of preschool. A condition for participating in the study was that children were full-time in preschool. The six preschools included 110 full-time children aged 4–6 years with valid accelerometer data, constituting a response rate of 90.2%. Two of the preschools served as a control group (33 children), while four preschools served as an intervention group (77 children). Due to the COVID-19 pandemic, we lost participation from one group of children in the control group.

The subjects were fully informed about the protocol prior to participating in the study. A written consent form was signed by the parents of the children according to accepted ethical research regulations. Approval to use the data and conduct the study was given by the Norwegian Centre for Research Data (NSD).

### 2.2. Intervention—Activity Videos

In the first week, both the control group and the intervention group carried out their ordinary activities. This was a normal week in the preschools, where they carried out the activities they usually engaged in according to their daily schedule. In the second week, the four intervention preschools used the activity videos, while the two preschools in the control group continued their ordinary activities. The activity videos included motion play, dance play, and strength and coordination exercises performed by children. “Active and Happy” is an online activity programme for schools and preschools. The videos for preschools were newly made, and none of the participating preschools had tested them before the activity measurements. The use of the videos is simple and requires no training. The activities can be performed without equipment except for a screen to view the videos. Children and employees together were supposed to copy the activities on screen by looking at the videos. The preschools were encouraged to use videos that provide a high heart rate for 20 min every day. The videos consisted of music and movement, including jumping, jumping jacks, high knee lifts, and activity trails, including running, jumping, and crawling over and under obstacles. The choice to use “Active and Happy” was made on the basis that the effects of the videos have not previously been evaluated.

### 2.3. Procedures

To answer the research question, accelerometers were used among preschool children for a period of two weeks. Accelerometers were chosen because they can detect the intensity, frequency, and duration of PA [42,43,44]. Moreover, the use of accelerometers made it possible to compare data with a national population study of PA level among preschoolers [42].

Accelerometer data were collected during February 2022. The pre-test was completed in the first week of the intervention period, while the post-test was carried out in the second week of the intervention period. Prior to signing the written consent form and the data collection form, parents received written information about the procedures and ethical standards for testing related to sports science. The Actigraph GT3X was utilized to objectively measure 4–6 year olds’ PA over five consecutive days in week one and five consecutive days in week two, which is more than the four days recommended by Migueles et al. [45]. The participants were instructed that the accelerometer had to be placed on the right hip, which is recommended [42,46], and worn every day during preschool except for during sleep, showering, or other activities involving water. The employees helped the preschool children place the accelerometers on their hips. Finally, the MVPA among preschool children and employees at preschool was categorized as 8:00 a.m.–3:29 p.m.

Raw data output produced from the accelerometers were expressed as counts per minute (CPM), which refers to all acceleration to which the accelerometer was exposed divided by the number of minutes the accelerometer was used [42]. According to the test protocol of Kolle et al. [42], counts were summed during 10 s intervals in order to capture as precise data as possible among the children and 60 s intervals among preschool employees. Furthermore, the accelerometer data were classified as sedentary, light, moderate and vigorous PA according to the divisions used in a national population study of PA levels among preschoolers [42]. In this study, the limit value for physical activity in MVPA is 2000 counts. According to international health recommendations, moderate and vigorous PA (MVPA) per day is the most relevant and used measure of the general PA level.

To initialize the accelerometers, download accelerometer data, and validate and create accelerometer data (MVPA), Actilife v6.13.3 (ActiGraph, LLC, Pensacola, FL, USA) was used. According to the test protocol, at least 480 min of daily recorded activity were required to obtain a valid day, and 20 min or more with consecutive zero counts were interpreted as non-wear time and removed [42]. Furthermore, according to Ato et al. [47], it is critical that the empirical data have a high degree of reliability, and in light of that, the preschool children were required to have at least two valid days each week to be included in the study [42,48,49].

### 2.4. Statistics

To examine changes in MVPA and CPM from pre-test to post-test, paired samples *t*-tests were used [50]. The level of significance was set at *p* < 0.05. The effect size was set with Cohen’s d, where 0.2 = small, 0.5 = medium, and 0.8 was large [51]. Statistical analyses were performed with SPSS, version 28.0 (IBM, Armonk, NY, USA).

## 3. Results

Figure 1 shows the activity level in CPM among preschool children in the control group and the intervention group at 4, 5 and 6 years old at both the pre-test and post-test.

The statistical analyses in Figure 1 show that there was no significant change (*p* > 0.005) in MVPA from pre-test to post-test among children in the control group and in the intervention group at 5 years of age. However, there was a significantly higher MVPA at post-test than at pre-test among 4 year old children (t_6_ = −6.1, *p* < 0.001, d = −2.31) and 6 year old children (t_55_ = −2.1, *p* = 0.037, d = −0.29).

Figure 2 shows the activity level of MVPA among preschool children in the control group and the intervention group at 4, 5, and 6 years old at both the pre-test and post-test.

The statistical analyses in Figure 2 show that there was no significant change (*p* > 0.005) in MVPA from pre-test to post-test among children in the control group and in the intervention group at 5 and 6 years of age. However, there was a significantly higher MVPA at the post-test than at the pre-test among 4 year old children (t_6_ = −4.8, *p* = 0.033, d = −1.05).

## 4. Discussion

The results show different effects of the activity videos according to the CPM and MVPA level among 4-, 5- and 6 year olds, with the greatest effects on the 4 year olds.

The first main finding was that the activity videos significantly increased CPM in preschool time among 4 year old children from pre-test to post-test with a large effect size [51] and also among 6 year old children, but at a borderline level close to 0.05 and with a small effect size. The difference between the pre-test and post-test among the children in the intervention group is also small (see Figure 1). It is appropriate to highlight that the control group had a significantly higher CPM at pre-test than both 4-, 5-, and 6year old children in the intervention group, and we argue that it is easier to increase CPM in the intervention group than in the control group. On the other hand, this difference disappeared during the intervention. Our findings indicate that the use of activity videos creates an increased general PA level during preschool time among preschool children. This is in line with Ramirez-Granizo [39], who found that activity videos can counteract sedentary behaviour and increase participation in physical activity. In addition, one study by Chacón-Cuberos [41] highlights that students have a positive attitude to physical activity using “exergames”. Our findings are in line with Peng et al. [41], who highlight that playing active video games generally is equivalent to light-to-moderate PA among children. Furthermore, Ramirez-Granizo et al. [39] found that activity videos are not able to replace physical activity in MVPA. However, the study of Kooiman et al. [37] shows that exergaming among secondary students can affect their heart rate, corresponding to moderate intensity activity. The children in our study are younger than in the study of Kooiman et al. [37]. We argue that our findings indicate that the 4 year olds especially found the activity videos interesting and funny. The activity videos may be a new experience for the youngest children, and this may have affected their PA level positively. The assumption that new experiences may influence the children’s intrinsic motivation is in line with Lohbeck et al. [52], who stated that a possible reason for such findings is that small children are typically more motivated for physical activities than older children. This highlights the importance of the employees as role models for both 4-, 5- and 6yearold children’s physical activity in preschool. This finding corresponds with Bugge and Froberg [24], who pointed out the importance of initiating or participating personally in physically active play that makes the children breathe hard. This may help to extend the duration of the physical activity. In combination with the fact that children find it fun to play with adults, children’s inner motivation can be stimulated by physically active play. In contrast, Vanderloo et al. [25] stated that a lack of stimulation or inactive role models demotivate children’s participation in physically active play. Positive experiences with physical activity can form the basis for a lifelong interest in being physically active [18,19,20]. With active employees, preschools can also help reduce sedentary time. In addition, a cross-sectional study reports that the total amount of PA decreased by an average of 4.2% each year from the age of 5 to 18 [53]. Furthermore, according to Lohbeck et al. [52], the significant mediating effect of intrinsic motivation suggests that children who feel competent in physical activities perform better when they are intrinsically motivated, that is, performing physical activities for their inherent pleasure without any external pressure. Preschools have to provide opportunities for ample physical activity. The use of activity videos can be an important contribution to offering varied physical activity and, as a result, make more children interested in physically active play.

The second main finding is that the 4 year olds were the only age group with significant effects regarding the use of videos in relation to MVPA, with a large effect size [51]. Preschool children play bodily, and the intensity and frequency of play increase significantly as the children age, reaching their highest point around 4–5 years of age [53]. This is in line with the fact that the 4 year old children in our study have a high level of PA. The physically active play of the oldest children in preschool is characterized by using large muscle groups with basic, natural gross motor movements such as crawling, walking, running, climbing, and balancing [54]. Several studies show that children’s physical activity level in MVPA increases from the age of three to the age of six [6,55,56]. However, studies have also pointed out that children’s PA level decreases from the age of three [57] and from the age of 5–6 [42,53]. Among the oldest children in preschool, this physically active play will often become more chaotic and noisy; e.g., rough-and-tumble play is an essential dimension of social play among the oldest children in preschool [54,58]. Basic movement play and rough-and-tumble play are physical activities where the children are likely to be breathless, hot and sweaty. In these games, a child needs both motor skills and endurance, and this will thus be an important contribution to children complying with the World Health Organization [4] recommendation of 60 min of PA in moderate to high intensity daily. Bearing this in mind, the explanation that the 4yearolds are most physically active may be due to the lack of rough-and-tumble play. Furthermore, this can explain why the findings are different for various ages. Changes to make the videos more attractive to the oldest children could be to include more active play in line with play preferences among 5- and 6 year olds, e.g., rough-and-tumble play. A study found a connection between rough-and-tumble play and PA level [59]. An alternative to increase the 5- and 6 year old children’s physical activity in MVPA could be to use more rough-and-tumble play. Furthermore, preschool staff being involved and making efforts to promote children’s PA greatly impacts the total PA level of children [22,23,24]. Positive encouragement and involvement by preschool employees are associated with higher physical activity levels among children. The employees can act as transmitters of enthusiasm. The content may be videos where children in pairs play wrestling games with the aim of putting their partner down on the floor, play boxing, and perform other activities without music. Furthermore, according to Hussain [27], it is important that the staff play a central role in both limiting and enabling progress in physically active play, generating and maintaining the children’s interest in further exploration. This is particularly important in rough-and-tumble play, as chaotic play can lead to real fighting if the children do not interpret the play signal correctly. Active role models may prevent the lack of help or concrete guidance in the play, as Karlsen and Lekhal [26] point out. In addition, another alternative regarding the PA in the videos can be to include more open movement solutions. This means that the children can choose how they want to solve the movement challenge—in other words, provide opportunities that enrich and extend children’s own thinking. This strategy will include all children with different degrees of motivation, interests, and motor skills, e.g., if the child does not want to dance, they can choose to be Spiderman or a magic doll. Furthermore, it is possible that children in the videos show different movements so that participating children can choose their movements. This prevents the skills from excluding children from the activity, and the activity videos may positively contribute to the fulfillment of the health recommendations of PA.

## 5. Strengths and Limitations of the Study

The present study possesses several advantages. It includes a large number of participants. Furthermore, different types and sizes of preschools were also included in the study as a consequence of being randomly selected, which yielded a representative sample. Accelerometers, as an objective measurement, decrease subjectivity [60] and eliminate bias, such as social desirability and recall problems [43]. Furthermore, several researchers have identified accelerometers as the optimal method to capture PA in free-living situations [45,61]. The Actigraph GT3X is validated and reliability-tested for measuring PA levels for children aged 0–5 [43,61,62,63,64]. Furthermore, the effects of using music in activity videos have previously been studied, though mostly in adults [41]. The present study also includes music activity videos among children.

Nevertheless, the present study is not without limitations. The study was carried out during the COVID-19 pandemic, which led to some drop-out among the control group. One out of two preschools in the control group had many children affected by COVID-19, which resulted in few children aged four. Furthermore, the control group consisted of only eight preschool employees. This may have affected the results. Finally, the intervention group of 4 year old children was small and consisted of only seven children.

## 6. Conclusions

The current study demonstrates two main findings. The first main finding was that the activity videos significantly increased CPM at preschool among both 4- and 6 year old children. The second main finding was that the 4 year olds were the only age group with significant effects regarding the use of videos in relation to PA in MVPA. We will argue that even if the use of activity videos among preschool children in preschool increased the PA level for some groups, the use of activity videos among preschool children in preschool has an even larger potential to create more PA. In light of these findings, we argue that the activity videos did not offer the children at the age of five and six enough intensity to increase PA in MVPA. However, our findings related to the 4 year old children (and partly the 5 year old children) indicate that activity videos can be an important contribution to more PA among preschool children at preschool. Further research regarding the effect of different types of activity videos on the PA level among 4- to 6 year old children is required.

## Figures and Tables

**Figure 1 sports-11-00056-f001:**
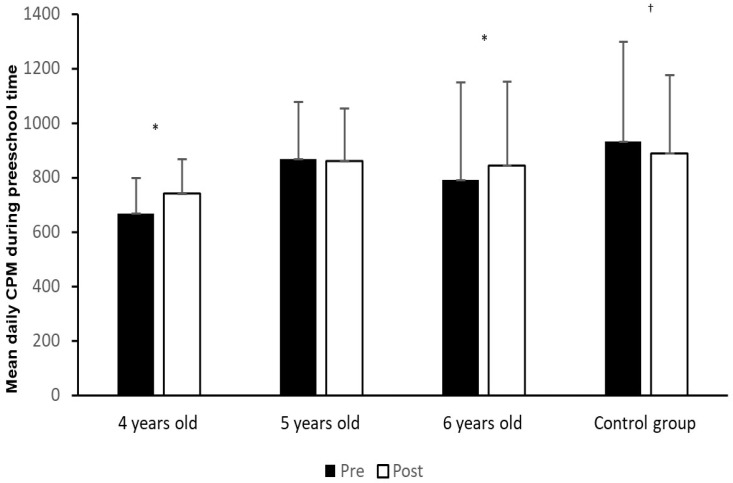
Activity level in CPM among preschool children in the control group and the intervention group at both pre-test and post-test. * Significantly higher CPM at post-test compared to pre-test in the intervention group (*p* < 0.05). ^†^ Significantly higher CPM in the control group compared to the intervention group.

**Figure 2 sports-11-00056-f002:**
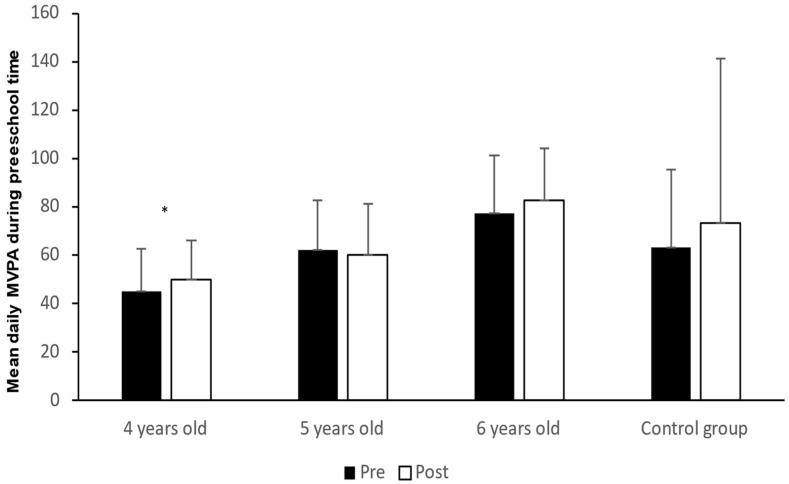
Activity level in MVPA among preschool children in the control group and the intervention group at 4, 5, and 6 years old in both pre-test and post-test. * Significantly higher MVPA at post-test compared to pre-test in the intervention group (*p* < 0.05).

## Data Availability

No data are available.

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
