# Peer review of "Activity Videos Effect on Four-, Five- and Six-Year-Olds’ Physical Activity Level in Preschool"

_sports, 2023, doi:10.3390/sports11030056_

Round 1

Reviewer 1 Report

The authors present a good research paper. 

  • The relevance of the topic: Good.
  • Introduction: Can be improved.
  • Methodology: Can be improved.
  • Results: Good.
  • Discussion: Good.  

However, ACCEPT AFTER MINOR REVISION. In general, the paper follows an adequate structure and correct scientific support and can be published considering some limitations. The study is interesting in the field of activity level in preschool time. However, there are a series of limitations that should be considered.

In the first place, carry out a review of the existing literature related to the subject, being essential to inquire into the MPDI – Sports journal itself, since there are papers related to its manuscript that can help to improve it. Therefore, include those references, if any, especially from the last five years. In addition, recommend reading some papers related to the topic of activity level in preschool time:

Dobell, A., Pringle, A., Faghy, M. A., & Roscoe, C. M. (2020). Fundamental movement skills and accelerometer-measured physical activity levels during early childhood: a systematic review. Children7(11), 224.

Mitchell, J. (2019). Physical inactivity in childhood from preschool to adolescence. ACSM's health & fitness journal23(5), 21-25.

Specific comments.

Title. The title of the manuscript is correct.

Abstract. Incorporate in the summary, a more precise sentence of the results.

Introduction. This section presents the problem in a coherent and clear manner with the correct support of the scientific literature. However, it is convenient to update the references, since there are different documents related to the subject and no mention is made, and it would even be interesting to mention the different existing studies related to activity level in preschool time. Also, it could be a future study of review. Some bibliographical references are attached to carry out the section of activity level in preschool time:

Aguilar-Farias, N., Toledo-Vargas, M., Miranda-Marquez, S., Cortinez-O’Ryan, A., Cristi-Montero, C., Rodriguez-Rodriguez, F., ... & del Pozo Cruz, B. (2021). Sociodemographic predictors of changes in physical activity, screen time, and sleep among toddlers and preschoolers in Chile during the COVID-19 pandemic. International journal of environmental research and public health18(1), 176.

Methods. Modify the method section, and specifically, in the section: Design.

-       Study design. To write the design section, we recommend that you take some of the following methodologists as references.

Ato, M., López-García, J. J., & Benavente, A. (2013). A classification system for research designs in psychology. Annals of Psychology29(3), 1038-1059.

Montero, I., & León, O.G. (2007). A guide for naming research studies in psychology. International Journal of Clinical and Health Psychology, 7(3), 847-862.

Results. Summary of study data and table are correct.

Discussion. The section Discussion is correct.

Conclusion. Differentiate the discussion of the main conclusions of the study. To do this, you must create this section. And modify the limitations of the study and locate them in said section at the end. Also, they must be direct, and highlight the main contributions of the study.

References. They should be reviewed and updated according to the publication standards. There are many errors in the references. Therefore, correct them and adapt them to the magazine's regulations.

Author Response

Dear Reviewer 1

Thank you for your comments.  We have rewritten the manuscript according to  the comments. 

Reviewer 2 Report

Introduction

- In the introduction, the authors mainly talk about the effects of physical activity on children's health and the impact of children's participation in physical activity on their future participation in adulthood. Also, the need to carry out interventions during childhood to increase physical activity has been raised. However, the main point that makes the introduction unacceptable is that different types of interventions that have been able to positively affect children's participation in physical activity have not been mentioned in previous research. In fact, it is not clear what kind of interventions have been able to increase physical activity in children. Therefore, the first point is that respected journalists should point to the types of interventions that had positive consequences.

- Also, more importantly, has video physical activity been used as an intervention in previous research? Basically, what is video physical activity? How has it been used and what are its consequences? The introduction lacks these contents and needs a general revision.

Method

- First paragraph of the method is not relevant here. Please remove it and place it in Instruments.

- Why these six schools were chosen? Were there any criteria?

- Why the number of participants in the intervention groups is more than control group? Please explain.

- Were there any criteria for inclusion and exclusion of the participants?

- Why you did choose “Active and Happy”?

- The procedure is really not a procedure. It is only related to accelerometer procedure. It is not obvious when was the posttest? Procedure in this form is not acceptable.

- What was the criteria for MVPA for this age? Please explain with reference?

- Please use ANCOVA for the comparisons.

Results

- Usually, the results should be begun with demographic data as well as descriptive data. However, there are no such data in the results.

- Lines 158-159 and 162-164: There are two captions for Figure 2?

- You have only measured MVPA? What about Light PA or sedentary time?

Discussion

- How did you know that the activity video is a new experience for children? Have you asked it before?

- Why are the findings different for various ages? Please explain.

Good luck

Author Response

Dear Reviewer 2

Thank you for your comments. We have rewritten the manuscript accordring to the comments (see the attached file).

Reviewer 3 Report

The abstract contains all the necessary information.

Introduction

The introduction is well-presented and developed.

Line 57: typo instead of -I contrast- you should have written -in contrast-.

What is the organized (sports or structured activities) physical activity that is included in the kids’ program during one day at school?

Was the goal to see if the videos were going to increase participation in self-organized physical activity only?

Methods

The methodology is described in detail.

My question regarding the methodology is if the kids used the accelerometers only at school.

It says that the employees helped the children place the accelerometers on their hips, but I am not sure when those were removed. Did they remove them before going home? You mentioned that they were not having them on during sleep, showering, or other activities involving water. It might be helpful to explain how many hours the children spend at school. Pre-school in our country starts at 7:30 am and they are out by 1:15 pm. They do not have time to nap etc.

Results

Include df in parenthesis next to your t value.

You may also want to calculate Cohen’s d to show the effect sizes.

Did you not compare the groups before the intervention? Based on the figures it seems as though the 5-year-olds were already having greater levels of physical activity (CPM at least) compared to the rest of the groups. This is just a suggestion for future studies. I am fine with the way the results are presented as the goal was to see the effect of the videos. It is, however, expected not to see much increase in the age groups that are already really active.

The discussion is clear and the strengths and limitations are mentioned.

Reference 5: no need to underline the references

Author Response

Dear Reviewer 3

Thank you for your comments. We have rewritten the manuscript accordring to the comments (see the attached file).

Round 2

Reviewer 2 Report

Thank you for your revision.

Best of luck.

Author Response

Thank you! The manuscript is uploaded in a new version.